# The fluid membrane determines mechanics of erythrocyte extracellular vesicles and is softened in hereditary spherocytosis

Daan Vorselen [1,2], Susan M. van Dommelen[3], Raya Sorkin[1], Melissa C. Piontek[4], Jürgen Schiller[5], Sander T. Döpp[1], Sander A.A. Kooijmans[3], Brigitte A. van Oirschot[3], Birgitta A. Versluijs[6], Marc B. Bierings[6], Richard van Wijk[3], Raymond M. Schiffelers[3], Gijs J.L. Wuite[1] & Wouter H. Roos[1,4]

Extracellular vesicles (EVs) are widely studied regarding their role in cell-to-cell communication and disease, as well as for applications as biomarkers or drug delivery vehicles. EVs contain membrane and intraluminal proteins, affecting their structure and thereby likely their functioning. Here, we use atomic force microscopy for mechanical characterization of erythrocyte, or red blood cell (RBC), EVs from healthy individuals and from patients with hereditary spherocytosis (HS) due to ankyrin deficiency. While these EVs are packed with proteins, their response to indentation resembles that of fluid liposomes lacking proteins. The bending modulus of RBC EVs of healthy donors is ~15 $k_{b}T$, similar to the RBC membrane. Surprisingly, whereas RBCs become more rigid in HS, patient EVs have a significantly (~40%) lower bending modulus than donor EVs. These results shed light on the mechanism and effects of EV budding and might explain the reported increase in vesiculation of RBCs in HS patients.

[1] Department of Physics and Astronomy and LaserLab, Vrije Universiteit Amsterdam, 1081 HV Amsterdam, The Netherlands. [2] Department of Oral Health Sciences, Academic Centre for Dentistry Amsterdam (ACTA), Research Institute MOVE, University of Amsterdam and Vrije Universiteit Amsterdam, 1081 LA Amsterdam, The Netherlands. [3] Department of Clinical Chemistry and Haematology, Division Laboratories, Pharmacy and Biomedical Genetics, University Medical Center Utrecht, Utrecht University, 3584 CX Utrecht, The Netherlands. [4] Moleculaire Biofysica, Zernike Instituut, Rijksuniversiteit Groningen, 9747 AG Groningen, The Netherlands. [5] Institute of Medical Physics and Biophysics, University of Leipzig, Medical Faculty, 04107 Leipzig, Germany. [6] Department for Stem Cell Transplantation, Princess Máxima Center for Pediatric Oncology & Wilhelmina's Children Hospital, University Medical Center Utrecht, 3584 EA Utrecht, The Netherlands. These authors contributed equally: Gijs J. L. Wuite, Wouter H. Roos. Correspondence and requests for materials should be addressed to G.J.L.W. (email: g.j.l.wuite@vu.nl) or to W.H.R. (email: w.h.roos@rug.nl)

Extracellular vesicles (EVs) are released by many cell types in vitro and in vivo and are present in most body fluids. They originate either from internal cellular organelles called multivesicular bodies (i.e., exosomes) or are shed directly from the plasma membrane (i.e., microvesicles)[1,2]. They are suggested to play a prominent role in cell-to-cell communication as inter-cellular transport vehicles carrying proteins and RNAs[3]. They have also been suggested to play a role in immune responses[4] and cancer progression[5,6]. EVs from red blood cells (erythrocytes, RBCs) are released both in vivo as well as under blood bank storage conditions[7]. A human red blood cell typically sheds 20% of its membrane area over its lifetime[8]. RBC EVs have been suggested to postpone the clearance of RBCs by the immune system[9], to play a role in blood clotting[10] and they have been put forward as a potential biomarker for dengue virus infections[11]. In patients with RBC infections such as malaria, the number of EVs is often elevated[12,13]. During malarial infection, RBC EVs were demonstrated to facilitate communication between malaria parasites[14,15].

RBC vesiculation is also relevant in blood disorders, for example in hereditary spherocytosis (HS). HS is one of the most common hereditary RBC disorders in the western world and is accompanied by increased release of EVs. This vesiculation is caused by a reduced linkage between the membrane and the underlying cytoskeletal spectrin network leading to loss of RBC membrane[12,16–19]. Reduced membrane surface area results in the formation of spherocytes. Spherocytes are less deformable, which causes these cells to be retained and cleared by the spleen. However, the relation between this blood disorder and EV release is still poorly understood as we are lacking structural and mechanical insights into EVs properties of both donors and patients.

These structural and mechanical properties of EVs influence their behavior, such as their interactions with cells[20–23]. Therefore, there has been a strong interest in quantifying EV mechanical properties[24,25]. At present, however, characterization of EVs is challenging due to their small size and a proper mechanical characterization of (RBC) EVs under physiological conditions has not yet been performed. Recently, we showed that the mechanics of small (<200 nm) synthetic vesicles, i.e., liposomes, are accurately described by a quantitative model based on Canham-Helfrich theory[26–29]. However, the mechanical properties of natural vesicles will result from a combination of lipidic contributions as well as that of membrane proteins and intraluminal proteins. The presence of such proteins could lead to a nonzero shear modulus of the natural vesicles, which might therefore be better described as thin elastic shells, and hence potentially show typical behavior thereof, such as buckling[30]. Moreover, thin shell behavior could be caused by spectrin structures, which is known to provide shear resistance to the RBC membrane[31–33]. A high percentage of the RBC membrane is occupied by membrane proteins (~20% at the hydrophobic core)[34]. The effect of these proteins on membrane mechanics is not entirely clear; nano-indentation studies with vesicles reconstructed from yeast membranes and influenza viruses both suggested that membrane proteins result in a large increase of the bending modulus of vesicles[35,36]. On the other hand, studies with model membranes have mostly reported membrane softening due to the presence of short peptides in the membrane[37]. Similar experimental studies with various larger membrane proteins showed either a neutral effect on membrane stiffness[38], or a decrease in the membrane stiffness[39]. Recent simulations reconcile some of these results by reporting how integral membrane proteins can have diverse effects on membrane mechanics, either softening, neutral or stiffening[40].

Here, we use atomic force microscopy (AFM) nano-indentation for mechanical characterization of RBC EVs. We use quantitative image analysis and show that RBC EVs remain in a rather spherical shape upon adhesion to the sample surface. Protein analysis and imaging of collapsed vesicles show that vesicles contain significant amounts of membrane-associated proteins. Interestingly, mechanically most EVs behave like empty liposomes with a fluid bilayer. We find that the bending modulus of healthy donor RBC EVs is ~15 $k_b T$, which is similar to previously reported values for liposomes[28] and which also corresponds well to bending moduli found in studies of the RBC membrane[41–43]. These results are compared to EVs derived from RBCs from patients with HS due to a mutation in the ANK1 gene. We reveal that these patient-derived EVs have an altered protein composition and a significantly softer membrane. This lower EV bending modulus could directly relate to the increased rate of vesiculation in HS patients.

## Results

**Characterization of red blood cell EV mechanics by AFM.** To study the mechanical properties of EVs, first the vesicles are imaged at high resolution to determine their geometry and center. RBCs derived from healthy volunteers were treated with a $Ca^{2+}$ iono-phore to stimulate EV formation. RBC EVs were subsequently attached to poly-L-lysine coated surfaces and imaged using atomic force microscopy (Fig. 1a). Due to electrostatic interactions with the surface, vesicles spread onto the surface. High resolution imaging revealed that EVs stay in a fairly spherical shape and have a homogeneous appearance. We applied a correction for the tip shape (Fig. 1b) and a correction for deformation due to applied imaging forces (see Materials and methods section). Subsequently we quantified the shape of EVs by measuring the height over the radius of curvature. For the 3 donor samples this ratio was (1–1.5) (Fig. 1c), indicating some spreading onto the surface. The difference in spreading indicates that there is some variation between the donors. Next, we calculated the initial radius of the vesicles, assuming surface area conservation, finding very similar values for all donors $88 \pm 2$ nm (standard error of the mean (s.e.m.), $N = 72$ vesicles), $88 \pm 3$ nm (s.e.m., $N = 55$) and $94 \pm 5$ nm (s.e.m., $N = 28$). (Fig. 1d). This is slightly higher than the average radius found during nanoparticle tracking analysis (NTA), which gave $R_0 = 71 \pm 1$ nm (s.e.m., 5 movies of 3 separate dilutions) (Fig. 1d, inset).

The mechanical behavior of the vesicles was studied by analyzing force indentation curves (FDCs), which were captured during indentation experiments. The curves were obtained by moving to the center of the EV and applying a force of 2–10 nN. Typically, FDCs revealed linear behavior and a subsequent flattening. Finally, a significant increase in force and two discontinuities can be observed, which correspond to the compression and penetration of both lipid bilayers (Fig. 2a). Some vesicles showed a much softer response, but with similar characteristics (Fig. 2b). A linear response is consistent with elastic behavior dominated by the physical properties of the membrane and indicates that the interior of the vesicle is not packed with polymerized proteins, in which case a Hertz-like behavior with a superlinear force-indentation response would be expected.

Sometimes large irreversible break events could be observed (Fig. 2c), which usually led to collapsed EVs. Previous reports on liposomes and natural vesicles have described a complete recovery after deep nanoindentations, with no detectable change in the geometry of the vesicles[35,36]. Here, we observed EV collapse in ~40% of cases for all donors. The shape of the non-collapsed EVs was either similar to before indentation or more flattened. Typical collapsed EVs are shown in Fig. 3. Some

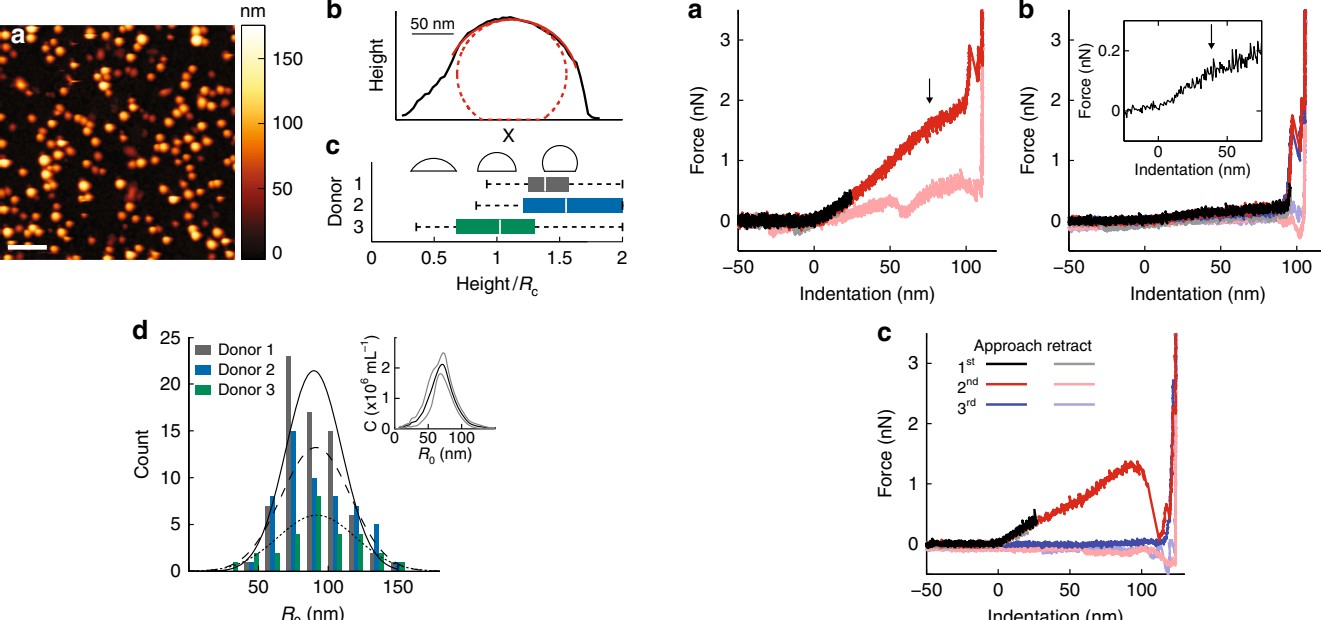

**Fig. 1** Geometry of adherent RBC EVs. **a** AFM topography image showing RBC EVs bound on a glass slide. Color scale indicates height. Scale bar length is 1 μm. **b** Line profile through slow axis of a single vesicle (in black). In solid red: the fitted spherical arc; in dashed red: the approximated vesicle shape after tip correction. **c** Average shape of vesicles defined by their height (H) over radius of curvature ($R_c$): 1.44 ± 0.03 (s.e.m., $N = 72$ vesicles), 1.52 ± 0.07 (s.e.m., $N = 55$) and 1.02 ± 0.08 (s.e.m., $N = 28$) for the three donors, respectively. Boxplots are shown in which the center white line indicates median, box limits indicate upper and lower quartiles and whiskers indicate 1.5× interquartile range. Reference shapes are shown in black for $H/R_c$ equals 0.5, 1, and 1.5. **d** Size distribution of vesicles. $R_0$ is the calculated radius of the vesicle while in solution. Black lines (solid, dashed and dotted for donor 1–3 respectively) show Gaussian fits. Inset shows the size distribution derived from NTA, where $C_i$ is the number concentration in particles per ml. Displayed is the mean (black line) ± s.d. (standard deviation) (gray lines) of three independent measurements. Mean radius of the vesicles is 71 ± 1 nm (s.e.m., 5 movies of 3 separate dilutions)

**Fig. 2** Typical FDCs on RBC EVs. **a** Two subsequent AFM force indentation curves (FDCs) showing an initial linear elastic response (black and red curves). The arrow marks a subtle flattening of the second FDC. Then an abrupt increase in stiffness occurs, followed by two break events, after which the glass surface (identified as a vertical line) is reached. The two break events correspond to the penetration of the lipid bilayers and the first one is typically larger than the second. Lighter colors indicate the mechanical behavior during AFM tip retraction. **b** Similar qualitative behavior as in **a**, but a much softer response. Inset shows smoothed data zoomed on the initial regime of the first indentation curve, where we can see that the initial linear response softens (black arrow). A third indentation curve is included revealing that this response is fully reversible (blue curve). **c** A FDC with a large discontinuity (red curve), after which the particle is ruptured (blue curve). All FDCs were obtained with EVs from donor 1

collapsed EVs appear as flat structures, with heights of 15–35 nm (Fig. 3a, Supplementary Fig. 1). Other EVs show elevated halo like edges, with similar maximum heights (Supplementary Fig. 1). Yet other EVs show partly elevated flat structures (Fig. 3b) or more complex structures (Fig. 3c, d). We compared the height of these structures with that of the lipid bilayer, which we measured when it was partly exposed at 4.1 ± 0.2 nm (s.e.m., $N = 9$ bilayers), indicating that the observed structures are much larger. In fact, the recorded structures resemble proteins and aggregates thereof observed on the RBC inner cell membrane[44,45]. Their height corresponds well to previous observations of proteins sticking out up to 10 nm above the inner membrane[44]. This suggests that the vesicles in Fig. 3b–d break at least partially open and expose their inner membrane, whereas the vesicle in Fig. 3a likely stayed intact. The ruptured EVs show that the membrane of RBC EVs contains a significant amount of proteins.

To investigate protein content, RBC ghosts and EV proteins were subjected to gel electrophoresis. RBC ghosts consist of the RBC membrane and proteins associated with the membrane. RBC ghosts can be isolated using hypotonic shock. In that way, hemoglobin levels are reduced dramatically and membrane associated proteins are enriched. In addition, RBC lysates were used to identify hemoglobin and other cytosolic proteins. RBC

proteins have been studied extensively, which makes protein identification possible without immune staining[46]. From electrophoresis we see that hemoglobin and band 3 are present in the EVs, as well as small amounts of protein 4.1 and 4.2 (Fig. 3e). In contrast to previous observations[47–49], we also find small amounts of spectrin. This is, however, not dependent on $Ca^{2+}$ stimulation, as we also find substantial spectrin levels in unstimulated RBC EVs (Supplementary Fig. 2). Actin however, seems to be absent in the EVs. Furthermore, we find that the membrane protein stomatin is enriched in EVs compared to RBCs, which agrees with previous findings[49]. This indicates that the EVs do have a distinct protein content compared to their donor cells, and that they do contain cytoskeletal elements and membrane proteins that might affect their mechanical properties. We also identified and quantified the lipids in RBC EVs, with thin layer chromatography (TLC) and matrix-assisted laser desorption and ionization mass spectrometry (MALDI-TOF MS). Using the method developed by Yao and Rastetter for TLC[50], we were able to separate and visualize seven (phospho)lipid classes. Analysis revealed that lipid composition of EVs was very similar to the native RBC membrane, with only a modest increase in phosphatidylserine (PS) in the vesicles (Supplementary Figs. 3, 4, Supplementary Data 1, 2).

**RBC EVs show mechanical behavior similar to fluid liposomes.** The mechanical properties of 153 EVs from three donors were

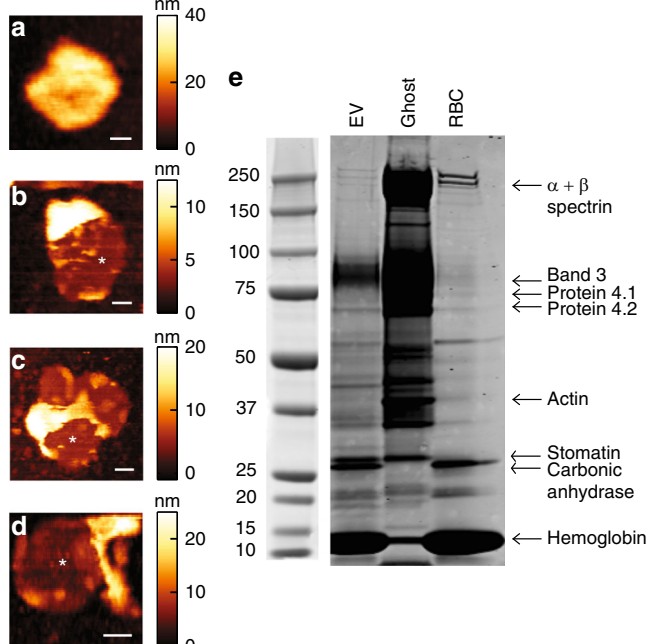

**Fig. 3** Pictures of collapsed EVs and their protein content. **a–d** AFM topography images showing collapsed EVs. Color scale indicates height. **a** Flat structure with mean height of about 22 nm. **b–d** Collapsed EVs exhibiting partly free bilayer (indicated with white asterisks). **b** Elevated part has mean height of about 26 nm. **c**, **d** Collapsed particles showing more complex structures. Scale bar length is 50 nm in all panels. **e** EV, ghost and RBC proteins were subjected to SDS-PAGE, by loading 10 μg protein per lane. After running, proteins were stained and protein patterns were compared with patterns known from literature. In this way, we were able to identify 9 well-known RBC proteins, which are differentially present in the three samples

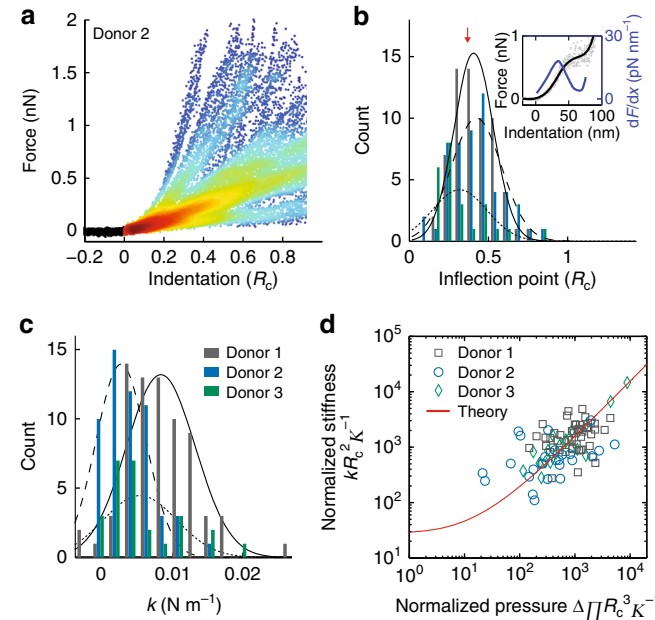

**Fig. 4** Mechanical characterization of RBC EVs. **a** Indentation behavior of 55 RBC EVs from donor 2 in a density plot. Colors indicate density of data points (blue and red indicate low and high density, respectively). Curves are shown until their first discontinuity. **b** Histogram of inflection points in the FDCs. Individual curves (gray dots show an example vesicle indentation in the inset) were smoothed (black curve, inset) and their derivative was taken (blue curve, inset). Main panel shows the location of the first peak of the derivative: $0.39 \pm 0.02\ R_c$ (s.e.m., $N = 64$ vesicles, in 8 cases no flattening was observed), $0.41 \pm 0.02\ R_c$ (s.e.m., $N = 53$, in 2 cases no flattening was observed) and $0.30 \pm 0.04\ R_c$ (s.e.m., $N = 24$, in 2 cases no flattening was observed) for the three donors, respectively. Black lines (solid, dashed and dotted for donor 1–3, respectively) show Gaussian fits. Red arrow indicates theoretically predicted value for a fluid vesicle. **c** Histogram of stiffness obtained by linearly fitting FDCs between 0.02–0.1 $R_c$. Lines represent Gaussian fits and line styles are equivalent to **b**. **d** Dimensionless pressure versus dimensionless stiffness. Theoretical prediction (solid red line) is based on an adaptation of Canham-Helfrich[26,27] theory to describe mechanics of small vesicles[28]. Data for donors were individually fitted to the theoretical prediction with the bending modulus κ as parameter. For visualization of the data for the three donors (different style markers) in this plot, the average κ (15) of the donors was used

analyzed (Fig. 4a). Recently, we investigated the mechanical behavior upon indentation of liposomes and found excellent correspondence with a Canham-Helfrich theory[26,27] based indentation model[28]. A signature of this behavior is a flattening of the FDC, marking the onset of formation of an inward directed lipid tether, and occurring at an indentation of 0.35–0.40 $R_c$, where $R_c$ is the radius of curvature of the vesicle. In contrast, in a thin elastic shell model, buckling is predicted to soften the response, which is expected to occur at smaller ($\sim$0.05 $R_c$) indentations for shells with the geometry of a vesicle[35]. We determined the inflection point of the FDCs of the donor samples from the peak in the derivative of smoothed FDCs (Fig. 4b, inset). The obtained distributions for the donor samples were centered at ~0.4 $R_c$, close to the previously predicted value by our quantitative model based on Canham-Helfrich theory (0.35–0.4 $R_c$)[28] (Fig. 4b). The good agreement with our model suggests that the bending behavior of the RBC EVs is dominated by a fluid membrane and that the membrane skeleton and membrane proteins are not resulting in a significant membrane shear modulus.

**Bending modulus estimation of RBC EVs.** Next, we set out to estimate the membrane bending modulus, an important intrinsic mechanical property of the membrane. We previously showed that the mechanical properties of adherent vesicles can be understood in terms of membrane bending and internal osmotic pressure, and that the bending modulus can be assessed using the vesicle stiffness, radius and tether force[28]. The EV stiffness was determined by fitting the initial linear response for indentations

up to 0.1 $R_c$ (Fig. 4c). There was some spread in stiffness between the donor samples; $10.9 \pm 0.5$ mN m$^{-1}$ (s.e.m., $N = 72$ vesicles), $5.8 \pm 0.4$ mN m$^{-1}$ (s.e.m., $N = 55$) and $8.2 \pm 0.9$ mN m$^{-1}$ (s.e.m., $N = 26$). Although we observe significant donor-to-donor variation in stiffness, stiffness is an extrinsic property that also depends on vesicle size and pressurization and does not necessarily reflect intrinsic differences between the donors. To estimate the pressure over the membrane, the retrace of indentation curves was analyzed. A tether, marked by a force plateau with force $F_t$, was detected during the retrace in ~60% of FDCs (donor 1: $F_t = 130 \pm 10$ pN, s.e.m., $N = 49$ tethers; donor 2: $F_t = 100 \pm 10$ pN, s.e.m., $N = 25$, donor 3: $F_t = 100 \pm 6$ pN, s.e.m., $N = 20$) (Supplementary Fig. 5). For our vesicles the adhesion-induced pressure is much larger than the additional indentation-induced pressure (Supplementary Fig. 6), in which case the tether force $F_t = 2\pi\sqrt{2\sigma\kappa}$, with σ the tension in the membrane and κ the bending modulus of the membrane[51–53]. We can subsequently estimate the pressure in the membrane using the Young-Laplace equation: $\Delta\Pi = 2\sigma/R_c$, with $\Delta\Pi$ the osmotic pressure difference over the membrane. With these measurements, and assuming

perfect vesicle osmometric behavior[54], we used our recent model to fit the bending modulus of the vesicles[28]. This revealed that there are no significant differences in bending moduli between the three donors (Supplementary Fig. 6), with average $\kappa$ at $15 \pm 1$ $k_bT$, (s.e.m., $N = 3$ donors) (Fig. 4d). Furthermore, we investigated EVs derived from non-stimulated RBCs to exclude significant effects of the $Ca^{2+}$ stimulation of the RBCs on the released EVs. This control resulted in a, within the error, similar bending modulus estimate of $17$ $k_bT$ (Supplementary Fig. 6).

**RBC EVs from hereditary spherocytosis patients are softened.** Finally, we compared the mechanical properties of RBC EVs from healthy donors with RBC EVs from patients with hereditary spherocytosis (HS). Dominantly inherited HS often is caused by mutations in the genes encoding ankyrin and band 3 (reviewed by Da Costa et al.[18]). In two of these patients (patient 1 & 2), HS is caused by heterozygosity for a novel 4 base pair insertion in *ANK1* (c.5201_5202insTCAG p.Thr1734fs). This 4 base pair insertion results in a shift of the reading frame, leading to a truncated ankyrin protein. One of these two patients underwent splenectomy (patient 2). The third patient (patient 3) shows heterozygosity for a novel nonsense mutation in *ANK1*: c.498C>G p.(Tyr166*). If stable, such proteins likely would be deficient in their function. Ankyrin truncation is expected to result in a disturbed cytoskeletal network and its connection to the plasma membrane. In turn this leads to increased vesiculation and hence a reduced ratio of surface area to volume, resulting in spherocytic cells (Fig. 5a). Such RBCs are poorly deformable and will be cleared prematurely from the blood circulation, leading to hemolytic anemia. Reduced RBC deformability was confirmed by laser diffraction ektacytometry (Fig. 5b).

The lipid and protein composition of patient RBC EVs were compared to that of healthy donor RBC EVs. Analysis of the phospholipid classes using TLC revealed a similar pattern in the donor and patient derived EVs, with the patient EVs having slightly more cholesterol and PS, and less phosphatidylethanolamine (PE) (Supplementary Fig. 3). To detect potential differences of specific lipid species we used matrix-assisted laser desorption and ionization mass spectrometry (MALDI-TOF MS). We found a mostly similar lipid composition between donor and patient derived EVs, with the only notable difference being a small shift in sphingomyelins (SM) towards species with a longer chain length in patient EVs (Supplementary Fig. 4, Supplementary Table 1, Supplementary Data 1, 2). Interestingly, we found reduced levels of α1-spectrin, ankyrin, and actin in patient EVs, while expression of these proteins is equal between healthy and patient RBCs (Fig. 5c). Tubulin levels, however, were increased in patient EVs, which reflects the increased expression of tubulin in the RBCs. The increased level of tubulin in the RBCs from the HS patient could be due to high reticulocyte numbers in this patient (~20%), which is a hallmark of HS indicating increased activity of the hematopoetic system trying to compensate for the hemolytic anemia[55]. However, reticulocyte derived EVs only represent a very small fraction (<0.1%) of the EV population, even in patient EVs, and hence cannot explain the increase in tubulin in the EVs (Supplementary Table 2). Overall, the lipid content of patient EVs and donor EVs was similar, whereas the protein content of the patient EVs had a distinct pattern when compared with the healthy donor sample (Supplementary Fig. 7).

We repeated the mechanical measurements for the HS derived EVs from the three patients. These EVs showed similar size ($R_0 = 84 \pm 5$ nm, s.e.m., $N = 3$ patients), and they appear similarly rounded as the donor samples with $H/R_c = 1.5 \pm 0.1$ nm (s.e.m., $N = 3$ patients). In about 50% of cases a membrane tether could be detected with average forces in the same range as the

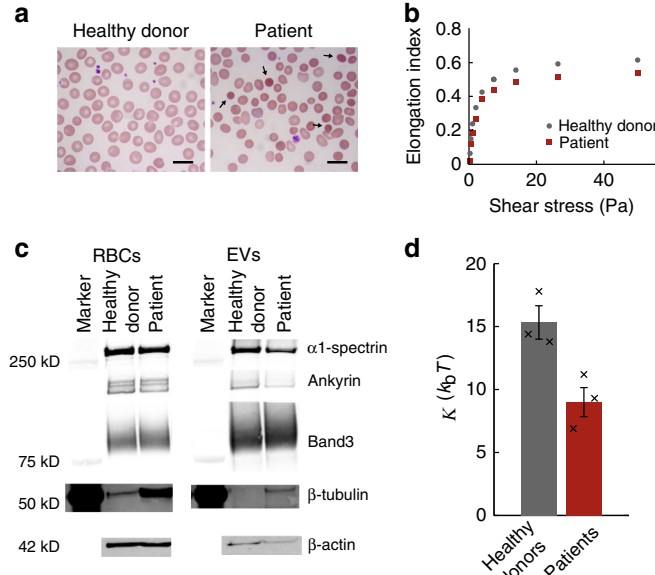

**Fig. 5** Characterization of RBCs and EVs derived from spherocytosis patients. **a** Blood smears stained with a May-Grünwald Giemsa stain. Black arrows show typical spherocytes. Scale bar length is 10 μm. **b** Elongation index of RBCs under increasing shear stress. Patient RBCs are less deformable under shear stress than RBCs from a healthy donor. **c** 30 μg RBC or EV protein was subjected to electrophoresis. After blotting the proteins, α1-spectin, ankyrin, band3, β-tubulin and β-actin were detected using immunoblotting. All patient data in a-c was from the patient with the 4 base pair insertion in *ANK1* that did not undergo splenectomy. Marker for β-actin is missing, as it was recorded at a different wavelength (700 nm) than the protein band (800 nm) (Supplementary Fig. 8). **d** Comparison of the bending modulus of EVs from the three donors and the three patients. Histogram bars indicate means, and error bars indicate standard errors (s.e.m.) of the 3 samples in each condition. Black crosses indicate bending moduli estimates for individual donor and patient samples. A two-sided *t*-test revealed that the difference between the donor and patient groups is statistically significant ($p = 0.02$)

donor samples (Supplementary Fig. 5). However, combining the stiffness, radius and tether force measurements to estimate the bending modulus for each patient (Supplementary Fig. 6) revealed that the bending modulus of patient derived EVs is $9 \pm 1$ $k_bT$ (s.e.m., $N = 3$ patients) (Fig. 5d). This is approximately 40% lower than EVs derived from the three healthy donors. Hence, HS patient derived EVs are significantly softened compared to healthy donor derived EVs.

## Discussion

In this study, we investigated the mechanical properties of EVs from RBCs. The large percentage of membrane area occupied by membrane proteins (~20% at the hydrophobic core), recently raised questions regarding the fluidity of natural membranes, both for synaptic vesicles[56] and the RBC membrane[34]. Perhaps surprisingly, we have shown that the mechanical behavior of RBC EVs agrees well with the theoretical behavior of a vesicle consisting of a fluid lipid bilayer without proteins. This is suggested by comparison of our experimental data with our recent model based on Canham-Helfrich theory[26,27] for indentation of small (<200 nm) fluid vesicles[28]. In particular, the measured inflection point in the FDCs corresponds well to the value predicted by our model (Fig. 4b), and the indentation data collapse on the predicted reduced pressure vs. stiffness curve (Fig. 4d). Furthermore, even the bending modulus of the EVs of the donors in this study

is similar to the bending modulus of liposomes with complex lipid mixture (designed to mimic the RBC lipid composition) obtained previously[28]. This suggests that the net effect of the various membrane-associated proteins in RBC EVs on the bending behavior of the membrane is neutral. This is different from the conclusions by Calo et al.[35], who suggested that membrane proteins have a strong influence on mechanics of small vesicles. However, their conclusions are based on the comparison between different studies, whereas our comparison is based on experiments conducted under the same conditions and using identical analysis.

Furthermore, this study illustrates the importance of pressurization due to deformation on the surface. The healthy donor samples have a stiffness that is up to a factor 2 apart, with $10.9 \pm 0.5$ mN m$^{-1}$ (s.e.m., $N = 72$ vesicles) and $5.8 \pm 0.4$ mN m$^{-1}$ (s.e.m., $N = 55$) for donor 1 and 2, respectively (Fig. 4c). Yet, taking into account the differences in spreading onto the surface and hence pressurization, we show that the intrinsic mechanical properties, as quantified by the bending modulus, are not significantly different (Supplementary Fig. 6). HS patient derived EVs have similar stiffnesses ($11 \pm 2$ mN m$^{-1}$ (s.e.m., $N = 3$ patients)), but here we demonstrated that the bending modulus of these vesicles is significantly lower.

For three healthy donor RBC EV samples, we found that the bending modulus is ~15 $k_bT$. Our estimate is comparable to the bending modulus found for the RBC membrane using flicker spectroscopy[41–43]. However the bending modulus of the RBC membrane has been estimated to be higher using micropipette aspiration and tether pulling experiments (~40 $k_bT$)[57,58], and optical tweezers assisted measurements of membrane fluctuations (~60 $k_bT$)[59]. The variability in the observed values for the RBC membrane makes it hard to directly compare the RBC EV bending modulus with that of the RBC membrane, but since the membrane modulus of healthy donors does not appear to be strongly affected by membrane proteins and the lipid composition of the EVs is comparable to the RBC membrane, the RBC membrane and RBC EV bending moduli are likely similar.

Interestingly, we find that EVs from patients with HS due to ankyrin deficiency show an approximately 40% lower bending modulus than donor derived samples. This is unlikely to be caused by differences in lipid composition, because we found only subtle differences in lipid composition between patient and donor derived EVs (Supplementary Fig. 4, Supplementary Table 1). Moreover, the small differences that were detected, include higher cholesterol content and a shift to longer chain length sphingomyelins in patient EVs. Both these observations, long chain length SM through thickening of the lipid bilayer and increasing the coupling between the two bilayer leaflets[60], would be more compatible with stiffening of the membrane than the observed softening[61]. We do, however, observe significant differences in donor and patient EV protein content. This makes it conceivable that EV proteins, rather than phospholipids, are likely to cause the ~40% decrease in the EV bending modulus in HS.

Our results also provide new insights into the pathophysiology and vesiculation process in HS. Suggested mechanisms for membrane shedding by RBCs are related to clustering of membrane proteins driving curvature generation[48,49,62] and to the balance between membrane bending and stretching of the spectrin cytoskeleton[62–64]. The weaker linkage of the membrane with the underlying cytoskeleton in HS likely causes loss of organization in the membrane. This is supported by the observation that diffusion of membrane proteins is faster in RBCs from patients with HS[65]. We speculate that loss of organization of the membrane gives room to accumulation of specific membrane proteins that lower the bending modulus locally and hence the energy barrier for vesicle formation. This would be consistent with the

observation that the bending modulus of spherocytic red blood cells is not decreased[43,58], and in this way, the loss of membrane linkage could lead to the reported increase in vesiculation in HS patients[12]. Ultimately, such increased vesiculation is the primary reason of the rounding and stiffening of HS RBCs, resulting in RBC clearance from the circulation and hemolytic anemia[12,16–18].

To conclude, we have presented an innovative approach to characterize the structural and material properties of extracellular vesicles. Surprisingly, the material properties of RBC EVs from healthy donors are similar to that of fluid liposomes. In contrast, the EVs from hereditary spherocytosis patients are significantly softer. This suggests a neutral effect for (membrane) proteins in donor EVs, whereas the different protein composition of patient EVs results in a lower bending modulus. The low rigidity of patient EVs fits with the augmented stiffness of HS patient RBCs and their increased vesiculation. Our approach has shed novel light on EV structure from healthy individuals as well as from HS patients, and is expected to be applied for the characterization and development of a variety of natural EVs and EV-based approaches in nanotechnology and nanomedicine.

## Methods

**Ethical statement.** The authors confirm that they complied with all relevant ethical regulations as they followed approved guidelines described in the University Medical Center Utrecht (UMC Utrecht) Biobank Regulations adopted by the Executive Board of UMC Utrecht. Informed consent was obtained from all participants.

**Blood smears.** Blood from healthy donors and a patient was collected in $K_2$EDTA (ethylenediaminetetraacetic acid) tubes. Patient blood was collected during regular controls at the outpatient clinic. Informed consent was obtained from all individuals, and procedures were performed in agreement with the declaration of Helsinki. Smears were prepared manually by spreading a drop of blood on a glass slide and were stained using May-Grünwald (J.T.Baker) and Giemsa (Merck) staining. Smears were imaged using an Axio Scope.A1 microscope (Zeiss).

**Red blood cell deformability.** Blood from healthy donors and the patient was collected in $K_2$EDTA (ethylenediaminetetraacetic acid) tubes. RBC deformability was analyzed using Laser-assisted Optical Rotational Cell Analyzer (Lorrca; Mechatronics, Hoorn, The Netherlands). In these measurements the elongation of the cells is measured under conditions of increasing shear stress. For this, blood samples were diluted 200 times in Iso-osmolar polyvinylpyrrolidone (PVP) solution (viscosity, 30 mPa s). One milliliter of the RBC in suspension was transferred into a static and rotating cylinder in the Lorrca analyzer and subjected fully automatically to a standardized increase of shear stress. The temperature was kept constant at 37 °C. Deformability was expressed as an elongation index (EI), as derived from the resulting ellipsoid diffraction pattern. The deformability curve was obtained by plotting the calculated values for the elongation index versus shear stress (Pa).

**Red blood cell stimulation and subsequent isolation of EVs.** Blood from healthy donors and the patients was collected in heparin tubes. PEGG elution columns (GE Healthcare) were filled with cellulose (1:1 w/w α-cellulose and cellulose type 50 in 0.9% NaCl). After washing the column with 0.9% NaCl, 4.5 ml of whole blood was applied on top of the cellulose. Columns were centrifuged for 5 min at 50×$g$, washed with 5 ml 0.9% NaCl, and centrifuged again to elute the RBCs. RBCs were washed with saline and resuspended in Ringer's buffer (32 mM HEPES, 125 mM NaCl, 5 mM KCl, 1 mM MgSO$_4$, 1 mM CaCl$_2$, 5 mM glucose, pH 7.4) to yield a final hematocrit of 40%. RBCs were stimulated with 4 μM Ca$^{2+}$ ionophore (A23187, Sigma) for 20–22 h, while tumbling at room temperature. RBCs were centrifuged for 10 min at 1000×$g$. Supernatant was diluted 10 times in phosphate buffered saline (PBS: 10 mM phosphate, 150 mM sodium chloride, pH 7.3–7.5, Sigma) and centrifuged again to remove residual RBCs. Large particles were depleted by centrifugation for 10 min at 10,000×$g$. Supernatant of 10,000×$g$ pellet was spun down for 70 min at 100,000×$g$ to pellet EVs. EVs were washed once in PBS. All EV isolation steps were performed at 4 °C.

**Ghost membrane preparation.** Washed RBCs were diluted 1:10 in hypotonic phosphate buffer (1.4 mM NaH$_2$PO$_4$, 5.7 mM Na$_2$HPO$_4$) supplemented with protease inhibitor cocktail (Roche) and were incubated for 2 h at 4 °C while gently tumbling. Ghost membranes were spun down at 43,000×$g$ for 10 min, without brake. Membranes were washed until the pellet was transparent and free of hemoglobin. Ghost membranes were resuspended in HEPES buffered saline (HBS, 10 mM HEPES, 150 mM NaCl, pH 7.4).

**Electrophoresis and immunoblotting**. Proteins were quantified using BCA (bicinchoninic acid) analysis (ThermoFisher Scientific) and equal protein amounts were subjected to gel electrophoresis, as indicated. Proteins were either blotted onto PVDF membranes (Merck Millipore) or stained in the gel using PageBlue (Life Technologies). For western blots the following antibodies were used: Anti-alpha 1 Spectrin (ab139403, Abcam), Anti-beta Tubulin (ab6046, Abcam), Anti-Band 3 (B9277, Sigma), Anti-β-actin (3700, Cell Signaling Technology), Anti-Ankyrin-1 (9473PA, IBGRL). All were used at 1000x dilution. Blots and gels were imaged using an Odyssey imager (LI-COR) after incubation with secondary antibodies (926–32211, 926–32212 LI-COR; A-21057, A-21076 ThermoFisher Scientific) at 5000x dilution. Uncropped images of all gels are presented in Supplementary Figure 8.

**Lipid extraction and quantification**. Lipids were extracted from the samples using the Bligh and Dyer method[66]. Samples were diluted seven times in a 2:1 methanol: chloroform (v/v) mixture. Samples were vortexed, after which chloroform and distilled water were added for a final ratio of 1:1:1 chloroform:methanol:water (v/v/v). Next, samples were vortexed and spun for 15 min at 4000×$g$ (4 °C). The bottom layer, containing the lipids, was collected and lipids were dried under nitrogen. Lipids were reconstituted in 2:1 chloroform:methanol (v/v). Phosphate was determined using the Rouser method[67]. Briefly, samples were dried by heating to 200 °C. 0.3 ml perchloric acid was added per sample and samples were heated to 200 °C for 45 min. Samples were cooled down to room temperature, 0.5 ml 1.25% hepta-ammoniummolybdate, 0.5 ml 5% ascorbic acid and 1.0 ml $H_2O$ were added per sample, and samples were reheated to 80 °C for 5 min. Absorbance was measured at 797 nm. A calibration curve of phosphate was used to interpolate the phosphate concentration in the samples.

**Thin-layer chromatography (TLC)**. TLC was performed according to Yao and Rastetter[50]. A TLC plate (silica on aluminum, Sigma) was washed with methanol and dried for 30 min at 150 °C. A full length predevelopment was performed in methyl acetate:1-propanol:chloroform:methanol:0.25%KCl (25:25:25:10:9, v/v/v/v/v) followed by drying for 30 min. 1.5 μg lipid per lane was applied onto the TLC plate. The TLC plate was developed halfway using the solvent used for predevelopment. The plate was dried and developed until the solvent front was about 1 cm beyond the end of the plate using hexane:diethyl ether:acetic acid (75:23:2). The plate was dried for another 30 min and finally totally developed using hexane. Detection was done by applying 10% copper sulfate hydrate in 8% phosphoric acid, followed by heating at 200 °C. Standard lipids were purchased from Lipoid.

**AFM experiments**. EVs were studied on poly-L-lysine coated glass slides in PBS. Slides were first cleaned in a 96% ethanol, 3% HCl solution for 10 minutes. Afterwards they were coated for 1 h in a 0.001% poly-L-lysine (Sigma) solution, rinsed with ultrapure water, and dried overnight at 37 °C. They were stored at 7 °C for a maximum of 1 month. For figure panel 1 A vesicles were attached to APTMS (Sigma) coated glass slides. After cleaning of the glass slides as indicated above, glasses were coated in 5 minutes in 0.2% APTMS solution (in ethanol). Slides were then stored in ethanol and rinsed with ultrapure water just before use. A 50 μL drop of vesicle solution was incubated on the glass slide. Vesicles were imaged in PeakForce Tapping™ mode on a Bruker Bioscope catalyst setup. Force set point during imaging was 100 pN–200 pN. Nano-indentations were performed by first making an image of a single particle, then indenting it until 0.5 nN and subsequently higher forces (2–10 nN) at a velocity of 250 nm s$^{-1}$ until the surface was reached. After indentation, typically another image was recorded to check for movement or collapse of the vesicle. Importantly, both before and after the vesicle indentation, the tip was checked for adherent lipid bilayers by pushing on the glass surface until a force of 5 nN. Tips used were silicon nitride tips with a nominal tip radius of 15 nm on a 0.1 N m$^{-1}$ cantilever by Olympus (OMCL-RC800PSA). Individual cantilevers were calibrated using thermal tuning.

**AFM image analysis**. Both images and force curves were processed using home-built MATLAB software. Size and shape were analyzed from line profiles through the maximum of the vesicle along the slow scanning axis. Circular arcs were fit to the part of the vesicle above half of the maximum height to obtain the radius of curvature, from which the tip radius (15 nm, as provided by the manufacturer) was subtracted. The height of vesicles was derived from FDCs, and the difference between the height obtained from FDCs and images was used for a subsequent correction of $R_c$ (Supplementary Fig. 9). For calculation of $R_0$ a minimum radius of curvature of 5 nm was assumed at the contact between the vesicle and the underlying surface, since a sharper contact angle would be non-physical[68].

**AFM FDC analysis**. Cantilever response was measured on the sample surface and fitted linearly. The resulting fit was subtracted from the measured response when indenting vesicles, to obtain FDCs. Contact point was found by using a change point algorithm[69] and occasionally manually adjusted. Before fitting, FDCs (each consisting of 10k data points) were smoothed (moving average with window length of 10 points). Stiffness of the liposomes was found by fitting a straight line in the interval between 0.02–0.1 $R_c$. To find the inflection point, FDCs were smoothed further (moving average with window length of ca. 40

points and Savitzky-Golay-filter with window length ca. 20 data points). Then, the derivative was taken numerically and the location of the maximum was obtained. For finding the tether force a step fitting algorithm based on the change point algorithm was used. Only clear force plateaus were included, and tethers with forces > 0.25 nN were excluded, since they could correspond to double bilayer tethers (Supplementary Fig. 5). Standard errors of the mean for tether forces were determined by 1000 bootstrapping repetitions. We note that the analysis of the pressure from tether pulling during the AFM tip retraction is only valid when the indentation-induced pressure is small compared to the initial adhesion-induced pressure. For a typical vesicle with dimensionless pressure 800, radius of curvature ($R_c$) 100 nm and bending modulus 15 $k_bT$, the initial pressure is ~45 kPa (pressure statistics on a per vesicle basis are presented in Supplementary Fig. 6). From our model, we estimate the vesicle volume change during indentation to be 0.3% for indentation 0.1 $R_c$ (which is the maximum indentation that we analyze for obtaining the vesicle stiffness). This leads to an additional pressurization of ~2 kPa, which is approximately 20-fold smaller than the initial pressure. For the dimensionless fit in Fig. 4d an interpolating function through 13 calculated theoretical pairs of values was created in Mathematica. The sum of the squared Euclidian distance between the logarithm of the resulting curve and the logarithm of individual data points was then minimized. Confidence intervals were estimated using the bias corrected percentile method with 500 bootstrapping repetitions, for which a set of observed value combinations equal in size to the original data set was randomly drawn and fitted.

**Nanoparticle tracking analysis (NTA)**. EVs were sized by recording 5 videos of 60 s using the NanoSight LM10 system (Malvern Instruments). A camera level of 11 was used and videos were recorded at 22 °C. Analysis of the videos was performed using the NTA 2.0 software, using default settings. Threshold was set at 5.

**Mass spectrometric analysis**. All samples were extracted prior to mass spectrometric characterization according to the method developed by Bligh and Dyer[66]. All MALDI mass spectra were recorded with DHB (0.5 M in methanol) as the matrix[70]. All lipid samples of interest were mixed 1:1 (v/v) with the matrix solution and subsequently applied onto the MALDI target under slight heating with a conventional hairdryer that helped to improve the homogeneity of crystallization.

All mass spectra were acquired on an Autoflex I MALDI mass spectrometer (Bruker Daltonics, Bremen, Germany) with ion reflector. The system utilizes a pulsed 50 Hz nitrogen laser, emitting at 337 nm. The extraction voltage was 20 kV and gated matrix suppression was applied to prevent the saturation of the detector by matrix ions. All spectra were acquired in the reflector mode using delayed extraction conditions.

Spectral mass resolutions, signal-to-noise (S/N) ratios, and peak intensities were determined by the instrument software Flex Analysis 3.0 (Bruker Daltonics). The mass spectrometer was calibrated using a lipid mixture of known composition.

Due to the small lipid amounts in the vesicle samples and their largely unknown compositions, no internal standards were used and this is the reason why only relative changes can be given, i.e., a lipid profiling approach is taken. Although the negative ion mass spectra were also recorded, changes of the (phospho)lipid compositions are most obviously reflected by the positive ion spectra.

**Code availability**. The custom (Matlab) code used for analyzing the data in this study is available from the corresponding authors upon reasonable request.

## Data availability

A reporting summary for this article is available as a Supplementary Information file. The data supporting the findings of this study are available from the corresponding authors on request.

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

## Acknowledgements

The authors would like to thank Tineke van Lingen for performing the thin-layer chromatography experiments and Fred MacKintosh for helpful discussions. D.V., W.H.R, and G.J.L.W. acknowledge support by the Netherlands Institute for Space Research (SRON, grant MG-10–07). W.H.R. acknowledges support via a NWO Vidi grant, the STW Cancer-ID program and a FOM projectruimte. The work of S.M.v.D. and R.M.S. on extracellular vesicles was supported by ERC starting grant 260627 'MINDS' in the FP7 Ideas program of the EU. R.S. acknowledges support through HFSP postdoctoral fellowship LT000419/2015. J.S. was supported by the German Research Council (SFB 1052/Z3).

## Author contributions

Conceptualization: D.V., S.M.v.D., R.M.S., G.J.L.W. and W.H.R; AFM data acquisition: D.V., R.S., M.C.P., S.T.D.; AFM data analysis: D.V., Mass Spectrometry: J.S., all other vesicle experiments and analysis: S.M.v.D., S.A.A.K., B.A.v.O.; patient sample acquisition: B.V., M.B.B., R.v.W.; software: D.V.; writing, original draft: D.V., S.v.D., G.J.L.W. and W.H.R.; writing, review editing: all authors; supervision: R.v.W., R.M.S., G.J.L.W. and W.H.R.

## Additional information

**Competing interests:** The authors declare no competing interests.

