## [Peer Review File · Nature Communications]

Reviewers' comments:

Reviewer #1 (Remarks to the Author):

The manuscript is interesting describing the important role of EV biomechanical properties.

However there are major issues outlined below.

Couple of points:

1. The conclusion that softer EVs secreted from RBC of HS disorder patients is based on findings from a single patient and hence not convincing. While this may be the case, the data certainly does not support this evidence since two healthy and one diseased patient sample was used.
2. And then the 2-10nN indentation force. This is a huge range and also Sharma et al. have shown in their 2010 ACS Nano article (doi: 10.1021/nn901824n) that exosomes were ruptured at 5nN.
3. There is very little if any characterization of these vesicles, besides AFM image and western blots. Do they have tetraspanins etc. is not known or not included.
4. Ghosts were treated with protease but not EVs. There is no discussion of the potential effects of treating the EVs with protease prior to mechanical measurements.

Reviewer #2 (Remarks to the Author):

Vorselena et al. report that the bending modulus of Extracellular Vesicles (EVs) derived from Hereditary Spherocytosis (HS) RBCs is ~50% lower than that in EVs from normal RBCs.

The authors treated RBCs donated by two healthy volunteers and one patient with hereditary spherocytosis with a Ca²⁺ ionophore to stimulate EV formation. This treatment would be expected to induce Ca²⁺ influx into the RBC leading to serious dysregulation of several cellular functions, and resulting in proteolysis, oxidation, irreversible shrinkage and phosphatidylserine (PS) exposure. The authors need to demonstrate that the Ca²⁺ ionophore released EVs are equivalent to EVs induced during circulation of normal RBCs. This is particularly relevant given that the protein analysis revealed the presence of a substantive level of spectrin (in contrast to previous studies). This suggests that the EVs released in this manner are not equivalent to naturally generated EVs.

As the authors point out, RBC loose about 20% of their surface area during aging in vivo. This results in a wide range of physical and biophysical properties of the RBC population. The data in Fig 2A,B indicate that there is also a large distribution of the physical properties of the EVs.

The authors found a substantive difference in the shear modulus between the EVs generated from the two healthy donors: 10.9 ± 0.5 mN/m (SEM, N = 72) and $178.5.8 \pm 0.4$ mN/m (SEM, N = 55). This again suggests substantive uncontrolled variability between different samples. More samples are needed to power the analysis.

The authors have not explained the reason for that the differences in distribution of values for the bending modulus for EVs from donors 1 and 2. It is not clear that it is valid to combine these data sets. A larger number of donors would be needed to support this treatment of the data.

The authors only examined one HS patient. Given the observed variability of the different parameters between different donors, this is not a sufficient sample size.

Moreover as the authors point out, a substantive level of tubulin was observed in the HS sample, suggestive of the presence of a significant fraction of reticulocytes. As reticulocytes mature rapidly in vitro, it is likely that a substantive proportion of the EVs are derived from reticulocytes rather than mature RBCs. These reticulocytes would need to be removed in order to make a direct comparison between HS and normal RBCs.

Further work is needed to support the author's conclusions.

Reviewer #3 (Remarks to the Author):

Vorselen and Roos
Nature Communications

This is an excellent paper involving measurements on exosomes generated from red blood cells. The measurements appear carefully done, including a characterization of the protein composition of the exosomes. The authors use a quantitative analysis developed by them and published elsewhere to provide meaningful insights into the properties of these structures and the effects of membrane proteins on the curvature elasticity of endosome vesicles. The validity of the approach using the Canham-Helfrich model of the membrane is well-supported by the data and represents an important strength of the work. I do have a few questions and concerns that should be addressed.

I do have a small problem with the discussion section at lines 282-290. The reason that cells from HS patients are less deformable than normal cells is their decreased surface to volume ratio. That much is consistent with the authors comments in other parts of the paper. However, the discussion that the bending stiffness of the membrane left behind might have a higher resistance to bending because of the removal of membrane with a lower bending stiffness, while possible, would have little to do with cell deformability because the membrane bending resistance involves much smaller energies than are involved with limitation on deformation caused by reduced surface to volume.

Furthermore, there is a report of bending stiffness measured by tether formation that yields a value of approximately 45 kT for the bending stiffness of RBC from mice deficient in band 3 (and spectrin) that are spherocytic. This is not significantly different from the value obtained for control RBC's using the same method (BJ 95:1826, 2008). These values are about twice as large as bending stiffness for lecithin bilayers measured by tether formation (reference 51 in the manuscript). This prior publication also contradicts the conclusion that HS membranes (at least on the cell) have lower bending stiffness than controls.

There is an important assumption in the analysis that the vesicle deformation can be modeled as the indentation by two probes from opposite directions. This approach neglects possible contributions from the adhesive interaction between the EV and the substrate. While I appreciate that this the energy of adhesion is unknown, making it a problem to incorporate it into the analysis, that does not mean that consideration should not be made of the potential effects that the vesicle-substrate interaction might have on the interpretation of the measurements. For

example, if there is reversible peeling of the membrane from the substrate at the boundary, that would add to the effective surface area of the vesicle and to the overall compliance of the EV during indentation. It seems like this could result in a lower-than-actual value for the calculated bending stiffness. Indeed the value of 15 KT is about three times smaller than other published values for the bending stiffness of red blood cells (BJ 95:1826, 2008).

The lack of knowledge of the adhesion energy between the vesicle and the substrate could be overcome by inducing adhesion by using excluded polymers instead of poly-lysine. The energies of adhesion resulting from flocculation forces have been well-characterized. Although it is possible these adhesion energies may be too low to stabilize the vesicle position on the surface, the ability to control the adhesive energy might enable critical testing of some of the assumptions made in the analytical framework.

Line 109: "The difference in spreading between the samples can be attributed to either surface preparation or variation between the two donors." This question seems easily settled by performing experiments on the same donor repeated on multiple substrates. How many different surfaces were used for each donor? I would think you would want at least three replicates, and more if the variability is high.

The theory assumes that the vesicle is a perfect osmometer, but red cells are known to behave as slightly imperfect osmometers with the expected change in volume accounting for 60 to 70% of that of a perfect osmometer. (See Ponder's book, Hemolysis and Related Phenomenon). This correction should be included for vesicles containing hemoglobin.

It is true that for normal red cell membranes, the position of the protein in the gel provides a reasonably good indication of the type of protein involved. But in membranes deficient in certain proteins, this method becomes less reliable because of proteins with similar molecular weights that, while less prevalent, may run at or near the same location. This is particularly true of the band 3 region, but is a concern for other bands as well. Some immunochemistry to confirm the presence or absence of key proteins is preferred.

While most of this is dealt with in their prior publication, I am a little confused about how the bending stiffness and pressure are determined. My understanding is that the initial FDC depends on both the pressure and the bending stiffness, and that the pressure is determined subsequently based on the force required to draw a tether out of the vesicle as the AFM probe is withdrawn. But the intra-vesicular pressure depends both on the initial pressure set by the strength of the adhesive interaction with the substrate, and hydrostatic pressures generated osmotically by the change in vesicle volume that occurs during indentation. What is unclear is whether a full energy minimization is done during the retraction phase when the pressure is being calculated. This seems essential because the deformation of the vesicle during retraction is different from the deformation during indentation, and therefore these two pressures are in general not the same. It may be that they are approximately the same because the osmotically-induced pressures are small compared to the initial adhesion-generated pressure, but the case for this is not well made.

Reviewer 1

“1. The conclusion that softer EVs secreted from RBC of HS disorder patients is based on findings from a single patient and hence not convincing. While this may be the case, the data certainly does not support this evidence since two healthy and one diseased patient sample was used.”

We agree with the reviewer that the results with only 1 patient may have been too preliminary. We performed experiments on 3 more samples: 1 additional donor, and 2 additional patients, making for a total of 3 donor and 3 patient samples. The results were consistent with our earlier measured samples. For the donor samples we found a bending modulus 15 ± 1 kT (s.e.m.), and for the 3 patient samples we found 9 ± 1 kT (s.e.m.). Using a two-sample t-test, we were able to show that this 40% decrease in bending modulus for the patient samples is statistically significant ($p \sim 0.02$). We updated the main text and the figures to present this additional data.

“2. And then the 2-10nN indentation force. This is a huge range and also Sharma et al. have shown in their 2010 ACS Nano article (doi: 10.1021/nn901824n) that exosomes were ruptured at 5nN.”

While we indeed briefly discuss vesicle rupture in our manuscript, the majority of our results (including the bending modulus estimates) are based on the analysis of the initial part of the indentation curve (up to $0.1 R_c$), in which the maximum force is typically ≤ 0.2 nN. Our experiments were not designed as a systematic investigation into vesicle collapse, which would likely depend on multiple factors next to the indentation force (e.g. AFM tip size, indentation speed). Such an investigation, while interesting, is to our belief beyond the scope of this manuscript.

We would also like to note that the mentioned work by Sharma *et al.* reports a study of vesicle rupture during imaging of dried vesicles *in air*, whereas our work is performed in near physiological conditions (our EVs remained in a liquid environment with physiological salts and pH at all times). The mechanical properties of vesicles are likely to be severely affected by drying them. (On top of this, in the work by Sharma *et al.* also ~ 100 fold more indentations are made at ~ 1000 fold higher speed than in our work, since they performed high force imaging as opposed to a few centered nanoindentations.) This makes a direct comparison between our and their results far from obvious. This is the reason that in the part of the manuscript where we discuss vesicle collapse, we only make the comparison with the results by Calo *et al.* (Nanoscale 2014, 6:2275-85) and Li *et al.* (Biophys. J. 2011, 100: 225-234).

“3. There is very little if any characterization of these vesicles, besides AFM image and western blots. Do they have tetraspanins etc. is not known or not included.”

The majority of this paper is based on characterization of the extracellular vesicles by AFM nanoindentation. This is largely because such a detailed mechanical analysis (in near-physiological conditions, and with analysis based on the Canham-Helfrich model) is novel and revealed new insight into the mechanical and structural behavior of RBC EVs. Importantly, the majority of our data is not obtained through AFM imaging (as the reviewer states), but through force spectroscopy measurements.

However, we agree with the reviewer that further characterization of the EVs than only with western blots is desirable. Therefore we included mass spectroscopy and thin layer chromatography for an in-depth analysis of the lipid composition of the EVs. We focused on lipid analysis, as the role of lipids on membrane mechanics (such as chain length, saturation degree and cholesterol content) is understood to a reasonable degree. For (membrane) proteins, much less is known how specific protein species affect membrane mechanics. This is largely illustrated by the fact that the membrane bending modulus of the red blood cell has been studied for decades (for example: Evans, Biophys J. 1983, 43:27-30), but to this day little is known regarding the contribution of specific proteins to the bending modulus of the red blood cell membrane. More generally, the effect of specific proteins and peptides on the membrane bending modulus - with some notable exceptions (mostly channel-forming antibacterial peptides) - is still largely unknown (Dimova, Adv. Coll. Interf. Science 2014, 208:225-234). To our knowledge, this also holds true for tetraspanins; whether/how they affect the membrane bending modulus is unknown.

We now mention this further characterization more prominently in the text. The mass spectrometry analysis appears earlier in the text, with our initial discussion of donor EV lipid composition (page 8), and we restructured our later discussion of these results (page 10).

4. Ghosts were treated with protease but not EVs. There is no discussion of the potential effects of treating the EVs with protease prior to mechanical measurements.

None of the mechanical measurements in this manuscript were performed after protease treatment. (neither any of the vesicle samples, nor the red blood cell deformability measurements).

Reviewer 2:

“The authors treated RBCs donated by two healthy volunteers and one patient with hereditary spherocytosis with a Ca²⁺ ionophore to stimulate EV formation. This treatment would be expected to induce Ca²⁺ influx into the RBC leading to serious dysregulation of several cellular functions, and resulting in proteolysis, oxidation, irreversible shrinkage and phosphatidylserine (PS) exposure. The authors need to demonstrate that the Ca²⁺ ionophore released EVs are equivalent to EVs induced during circulation of normal RBCs. This is particularly relevant given that the protein analysis revealed the presence of a substantive level of spectrin (in contrast to previous studies). This suggests that the EVs released in this manner are not equivalent to naturally generated EVs.”

To better represent the potential effect of stimulation with Ca^{2+} ionophore we now include a comparison of stimulated vs non-stimulated EV proteins by western blot. Importantly, this revealed substantial spectrin levels in non-stimulated EVs. We included a supplementary figure (Fig S2) to show this data, and we refer to this figure in the main text where we mention the presence of spectrin in our vesicles.

Due to experimental limitations, we need to perform some type of stimulation. This is especially true for the patient samples, where we would otherwise have too little material to perform the mechanical characterization by AFM. However, if possible a control would indeed be essential and for the donor samples this was possible. We performed such a control experiment with unstimulated vesicles, which revealed that the bending modulus of unstimulated vesicles is similar to stimulated vesicles. This control is presented in figure S6. While we agree that there indeed could be differences between EVs generated by stimulation and EVs obtained without stimulation, this important control shows that the mechanical properties are the same and the mechanical characterization is exactly the focus of our manuscript.

We also like to point out that, to our knowledge, most previous studies made use of some sort of stimulation that will affect the RBCs and potentially the excreted EVs. For example, spectrin free vesicles were observed by Lutz *et al* (J. Cell. Biol. 1977, 73: 548-560) and Muller *et al* (Biochim. Biophys. Acta 1981, 649:462:470) but only under ATP-depletion; by Hagerstrand *et al* (Biochim. Biophys. Acta 1994, 1190: 409-415), who stimulated using Ampiphile, by the Jong *et al* (Biochim. Biophys. Acta 1996, 22:101-110) during long term storage under ATP-depletion.

“As the authors point out, RBC loose about 20% of their surface area during aging in vivo. This results in a wide range of physical and biophysical properties of the RBC population. The data in Fig 2A,B indicate that there is also a large distribution of the physical properties of the EVs.”

Indeed there is variability in the indentation behavior within the vesicle population, as shown in figure 2A,B. However, we note that this does not automatically mean that there is a large variation in the mechanical properties of the EVs. Variations in particle-specific properties, such as size and degree of spreading onto the surface, will certainly lead to variations in the observed indentation behavior. Therefor a size and absorption geometry independent parameter needs to be found to describe the mechanics of each vesicle. This is exactly what we do, as we outline in the following: The theoretical prediction is that the vesicle stiffness, when normalized by κ/R_c^2 , should only depend on the pressure, radius and bending modulus, and in fact, only through the very specific relation $\Delta\pi \cdot R_c^3/\kappa$. By plotting the data as we do in figure 4D and observing a collapse of the data (both within the data measured for each donor, and in between the donors) onto the theoretical curve, we have tested this qualitative prediction. Hence, the bending modulus (as the most important intrinsic measure of the mechanical properties) of these vesicles is similar, and the observed vesicle-to-vesicle variation can be understood in terms of variation in $\Delta\pi \cdot R_c^3/\kappa$ (vesicle radius and spread, in the specific combination implied by that

relationship). We now make explicit in the text that variations in stiffness do not directly inform us on differences in physical properties, by stating:

“Although we observe significant donor-to-donor variation in stiffness, stiffness is an extrinsic property that also depends on vesicle size and pressurization and does not necessarily reflect intrinsic differences between the donors.”

and we explicitly mention that the intrinsic property, the bending modulus, is similar between the vesicles, by stating:

“This revealed that there are no significant differences in bending moduli between the three donors”

“The authors found a substantive difference in the shear modulus between the EVs generated from the two healthy donors: 10.9 ± 0.5 mN/m (SEM, N = 72) and $178.5.8 \pm 0.4$ mN/m (SEM, N = 55). This again suggests substantive uncontrolled variability between different samples. More samples are needed to power the analysis.”

In accordance with the reviewer’s suggestion, we added three additional samples: 1 additional donor, and 2 additional patients, making a total of 3 donor and 3 patient samples. The results were consistent with our earlier measured samples. For the donor samples we found a bending modulus 15 ± 1 kT (s.e.m.), and for the 3 patient samples we found 9 ± 1 kT (s.e.m.). Using a two-sample t-test, we were able to show that this 40% decrease in bending modulus for the patient samples is statistically significant ($p \sim 0.02$). We updated the main text, and the figures, to present this additional data.

In addition, we note that we do not measure a shear modulus. In fact, we make a point in the paper that vesicles can be described merely by their fluidic membrane, and therefore do not have a measurable shear modulus. The values that are mentioned here by the reviewer (10.9 ± 0.5 mN/m (SEM, N = 72) and 5.8 ± 0.4 mN/m (SEM, N = 55)) are corresponding to vesicle stiffness. Stiffness is an extrinsic property of the vesicles and as now also better explained in the manuscript (see also above), depends also on the vesicle geometry (and pressurization). Hence, variation in stiffness does not necessarily correspond to variation of intrinsic vesicle properties. In the manuscript, we perform an analysis that takes into account stiffness, shape and pressurization and we derive an intrinsic mechanical property of the membrane: the bending modulus (Vorselen *et al*, ACS Nano 2017, 11; 2628:2636). The bending modulus of the three donor samples are very similar and do not differ significantly: $\kappa = 15 \pm 1$ kT (s.e.m.) $k_b T$. As also stated in our reply to the previous comment of the reviewer, we adapted the main text to convey the message more clearly.

“The authors have not explained the reason for that the differences in distribution of values for the bending modulus for EVs from donors 1 and 2. It is not clear that it is valid to combine these data sets. A larger number of donors would be needed to support this treatment of the data.”

We agree that combining those data sets can raise some questions. With the addition of a third donor sample, we no longer pool the data. However, as we explain above, there is only significant spread between the donor samples in extrinsic properties, and we do not observe significant differences in bending moduli between donor samples. As also stated above, we now make this point clearer in the manuscript.

The authors only examined one HS patient. Given the observed variability of the different parameters between different donors, this is not a sufficient sample size.

We agree with the reviewer that this is not a sufficient sample size. As described above, we have added three additional samples. We do not observe significant variability of the bending modulus between the donor samples and we find a statistically significant difference ($p \sim 0.02$) between donor and patient samples.

“Moreover as the authors point out, a substantive level of tubulin was observed in the HS sample, suggestive of the presence of a significant fraction of reticulocytes. As reticulocytes mature rapidly in vitro, it is likely that a substantive proportion of the EVs are derived from reticulocytes rather than mature RBCs. These reticulocytes would need to be removed in order to make a direct comparison between HS and normal RBCs.”

To address this valid point of the reviewer, we analyzed vesiculation rates of reticulocytes and mature red blood cells after ionophore stimulation and present the data in the new supplementary Table S2. Using a Percoll gradient, we were able to separate reticulocytes from the mature red blood cells of a healthy donor. Using nanoparticle tracking analysis, we showed that reticulocytes release around 300 times less EVs than mature RBCs. Although HS patients have higher reticulocyte counts (around 15-20%), this means that in the patient material still only one out of every 1500 EVs is reticulocyte-derived. Moreover, patient 2 had a splenectomy, which led to lower reticulocyte counts, in the range of healthy donors (2-3%), which shows that the effect we see of HS on the mechanical properties of EVs is not due to higher reticulocyte counts in HS.

“Further work is needed to support the author’s conclusions.”

We agree and we have performed this requested further work in order to provide a good basis to support our conclusions.

Reviewer 3:

“This is an excellent paper involving measurements on exosomes generated from red blood cells. The measurements appear carefully done, including a characterization of the protein composition of the exosomes. The authors use a quantitative analysis developed by them and published elsewhere to provide meaningful insights into the properties of these structures and the effects of membrane proteins on the curvature elasticity of endosome vesicles. The validity of the approach using the Canham-Helfrich model of the membrane is well-supported by the data and represents an important strength of the work. I do have a few questions and concerns that should be addressed.”

We thank the reviewer for the enthusiastic comments about our work.

“I do have a small problem with the discussion section at lines 282-290. The reason that cells from HS patients are less deformable than normal cells is their decreased surface to volume ratio. That much is consistent with the authors comments in other parts of the paper. However, the discussion that the bending stiffness of the membrane left behind might have a higher resistance to bending because of the removal of membrane with a lower bending stiffness, while possible, would have little to do with cell deformability because the membrane bending resistance involves much smaller energies than are involved with limitation on deformation caused by reduced surface to volume.”

We agree with the reviewer, and we removed this part of our discussion.

“Furthermore, there is a report of bending stiffness measured by tether formation that yields a value of approximately 45 kT for the bending stiffness of RBC from mice deficient in band 3 (and spectrin) that are spherocytic. This is not significantly different from the value obtained for control RBC's using the same method (BJ 95:1826, 2008). These values are about twice as large as bending stiffness for lecithin bilayers measured by tether formation (reference 51 in the manuscript). This prior publication also contradicts the conclusion that HS membranes (at least on the cell) have lower bending stiffness than controls.”

We thank the reviewer for pointing out this article. We included it in our discussion and changed our statement that the bending modulus of spherocytic red blood cells is higher than for healthy RBCs accordingly. Instead, we now state that the bending modulus of spherocytic cells is not lower than healthy RBCs.

“There is an important assumption in the analysis that the vesicle deformation can be modeled as the indentation by two probes from opposite directions. This approach neglects possible contributions from the adhesive interaction between the EV and the substrate. While I appreciate that this the energy of adhesion is unknown, making it a problem to incorporated it into the analysis, that does not mean that

consideration should not be made of the potential effects that the vesicle-substrate interaction might have on the interpretation of the measurements. For example, if there is reversible peeling of the membrane from the substrate at the boundary, that would add to the effective surface area of the vesicle and to the overall compliance of the EV during indentation. It seems like this could result in a lower-than-actual value for the calculated bending stiffness. Indeed the value of 15 KT is about three times smaller than other published values for the bending stiffness of red blood cells (BJ 95:1826, 2008)."

We agree that for large indentations surface effects, like peeling from the surface could affect our experiments. However, we point out that we only use small indentations (up to $0.1 R_c$) for measuring the vesicle stiffness, which lies at the basis for our bending modulus determination. As we showed in previous work, primarily local deformations are expected at indentations of this size, definitely up to $0.1 R_c$ (Vorselen, ACS Nano 11, 2628). The broadening of the radius in the equatorial plane for such indentations is indeed small ($\sim 0.02 R_c$). Therefore, we believe it unlikely that at this point vesicle-substrate interactions significantly affect the observed force response. For deeper indentations, we would think that such effects indeed become important. Regarding our bending modulus estimate for RBC derived vesicles, there have indeed higher values for the bending modulus of RBCs been reported in literature. However, overall a broad range in bending moduli of RBCs has been reported. We do agree with the reviewer that we should have stated matters more carefully. We now mention in our discussion that our estimate of the vesicle membrane is on the lower side of the range (Evans, BJ 43, 27- 1983; Butler BJ 95, 1826- 2008; Betz, PNAS 106, 15320- 2009). However, some estimates of the RBC membrane bending modulus that have been made are in the same range as ours (Duwe, Phys 51,945-961, Brochard; J.Phys 36, 1035-1047; Park, PNAS 107, 6731-6736). Moreover, we note that excreted vesicles do not necessarily have the same bending modulus as the red blood cell membrane.

"The lack of knowledge of the adhesion energy between the vesicle and the substrate could be overcome by inducing adhesion by using excluded polymers instead of poly-lysine. The energies of adhesion resulting from flocculation forces have been well-characterized. Although it is possible these adhesion energies may be too low to stabilize the vesicle position on the surface, the ability to control the adhesive energy might enable critical testing of some of the assumptions made in the analytical framework."

It is a very interesting suggestion to use excluded polymers to gain more experimental control over the vesicle surface adhesion. The addition of polymers to cell solutions has been used to induce cell-cell fusion and the adhesion energy of certain types of bilayers in the presence of polymers has been determined. However, in our experimental approach we are critically dependent on a firm enough adhesion of the vesicles to the surface in order to image them by AFM and to measure their mechanics. So, for low amounts of excluded polymers we expect poor vesicle adhesion. For high amounts of polymer there may be strong enough adhesion, but the resulting osmotic changes due to the presence of high concentrations of polymer is expected to influence the vesicle properties significantly. In addition, performing AFM experiments in solutions with high polymer concentrations is challenging and sometimes impossible. Although we haven't tried using excluded polymers, we have tried several other approaches for vesicle surface attachment (various concentrations of poly-l-lysine, bare glass, glass

treated with various silanes, mica) and a variety of problems were encountered. Optimizing such conditions is always very time-consuming. Overall in our AFM based approach we see considerable challenges in pursuing the suggestion of the reviewer. However, it is a very interesting idea and we believe that this certainly would warrant an entirely new study on its own.

“Line 109: “The difference in spreading between the samples can be attributed to either surface preparation or variation between the two donors.” This question seems easily settled by performing experiments on the same donor repeated on multiple substrates. How many different surfaces were used for each donor? I would think you would want at least three replicates, and more if the variability is high.”

For each donor sample many (> 10) different substrates were used. The spread between the vesicles on a single surface is far larger than the spread from surface to surface. So, we consider it more likely to be sample to sample variation than surface to surface preparation. We adapted this sentence in the manuscript accordingly.

“The theory assumes that the vesicle is a perfect osmometer, but red cells are known to behave as slightly imperfect osmometers with the expected change in volume accounting for 60 to 70% of that of a perfect osmometer. (See Ponder’s book, Hemolysis and Related Phenomenon). This correction should be included for vesicles containing hemoglobin.”

The intra-vesicular pressure depends both on the pressure set by the adhesive interaction as on the pressure resulting from the change in volume during indentation. We note that for estimation of the adhesion-induced pressure, which is much (~20 fold) larger than the indentation-induced pressure and therefore the dominant pressure (see below, in our response to the last comment to the reviewer), we do not make the assumption that the vesicle is a perfect osmometer, as we estimate the pressure from the tether force and not based on the vesicle volume change. For the much smaller indentation-induced pressure, one needs to decide if the RBC extracellular vesicles are perfect or imperfect osmometers. While vesicles reconstituted from the RBC membrane have been reported to act as perfect osmometers (Carruthers and Melchior, Biochemistry 1983, 22, 5797), it is indeed the contrary for RBCs (Ponder’s book and Pafundo et al. J. Biol. Chem., 2010, 6134, 44). To our knowledge, the situation for RBC EVs is not known. They could behave similar as RBCs, but could also act as perfect osmometers, or anything in between. As this is unclear, we have chosen to assume the EVs are perfect osmometers. We now clearly state that we make this assumption, with a reference, in the manuscript.

“It is true that for normal red cell membranes, the position of the protein in the gel provides a reasonably good indication of the type of protein involved. But in membranes deficient in certain proteins, this method becomes less reliable because of proteins with similar molecular weights that, while less prevalent, may run at or near the same location. This is particularly true of the band 3 region, but is a

concern for other bands as well. Some immunochemistry to confirm the presence or absence of key proteins is preferred.”

We agree with the reviewer that immunoblotting is a more powerful approach for identifying proteins than identification based on the position of the protein in a gel. However, the method that we have used is also well established (Costa *et al.*, NEJM 323:1046, 1990). Moreover, our main claim from these results is the following: “This indicates that the EVs do have a distinct protein content compared to their donor cells, and that they do contain cytoskeletal elements and membrane proteins that might play a role in their mechanical properties”. We believe that this claim is strongly enough supported by the current data. Finally, we did perform immunochemistry on key proteins, and we present the data in figure 5c.

“While most of this is dealt with in their prior publication, I am a little confused about how the bending stiffness and pressure are determined. My understanding is that the initial FDC depends on both the pressure and the bending stiffness, and that the pressure is determined subsequently based on the force required to draw a tether out of the vesicle as the AFM probe is withdrawn. But the intra-vesicular pressure depends both on the initial pressure set by the strength of the adhesive interaction with the substrate, and hydrostatic pressures generated osmotically by the change in vesicle volume that occurs during indentation. What is unclear is whether a full energy minimization is done during the retraction phase when the pressure is being calculated. This seems essential because the deformation of the vesicle during retraction is different from the deformation during indentation, and therefore these two pressures are in general not the same. It may be that they are approximately the same because the osmotically-induced pressures are small compared to the initial adhesion-generated pressure, but the case for this is not well made.”

The reviewer’s understanding of how the bending stiffness and pressure are determined are correct and the intra-vesicular pressure indeed depends both on the initial pressure set by the adhesive interaction and on the change in volume during indentation. To address the reviewer’s comment, we quantified the volume change occurring in our model during indentation. For small indentations, the volume change and increase in pressure are small: for an indentation of $0.1 R_c$ (the indentation we use for measuring the particle stiffness) the corresponding volume change is 0.3%. This corresponds to an increase in pressure of approximately ~ 2 kPa. This is small compared to the initial adhesion-generated pressure, which for a typical vesicle with dimensionless pressure 800, R_c 100 nm and bending modulus 15 kT is ~ 45 kPa. Hence, the adhesion-generated pressure is ~ 20 fold larger than the additional pressure caused by this indentation. We now present this data and argument in the supplementary discussion and we refer to it in the main text where we first discuss the bending modulus derivation.

Reviewers' comments:

Reviewer #1 (Remarks to the Author):

The authors have indeed increased the number of samples, marginally. I find the experiments were carefully done and also the analysis is quite nice. However, the effect is quite small between healthy and disease state which makes the conclusions a little bit uncertain due to possible artifacts. (40% decrease in bending modulus for the patient samples is statistically significant ($p \sim 0.02$)). Despite the somewhat limited number of sample sets and the broad distribution of stiffnesses, I am inclined to recommend a favorable editorial decision. Exosomes are of intense interest and there is a need for characterization and some standards in dealing with them. However, there is no doubt that they are a new biomarker for disease and have a bright future in research.

Reviewer #2 (Remarks to the Author):

The authors have addressed my comments and questions.

Reviewer #3 (Remarks to the Author):

My prior concerns have been well-addressed. I just have a few minor corrections and one request.

In Fig 2A the legend indicates three indentations, but only two are visible. Is the third one obscured or does the third color only apply to panels b and c?

I have no idea what is meant by "exponent a of the FDC 1.5," line 128.

Line 238 should refer to Fig 5, not Fig 4, and in line 239, it should refer to Fig. 6.

In supplemental Fig. 6A, does ANOVA support the conclusion that all of the patients were significantly different from the healthy donors?

Line 267: Supplemental Figure 6

Line 409. There is no Figure 5E.

Given the importance of the assumption that the pressure due to adhesion must be large compared with the pressure increases for small indentations it would be useful to report distribution of the pressures determined from the tethering forces shown in Supplemental figure 5. An example calculation is given in the Supplementary Discussion, but it is not obvious that all of the pressures measured from tethering forces are that high. A simple histogram of the values included as supplemental material would suffice.

Reviewer #3 (Remarks to the Author):

My prior concerns have been well-addressed. I just have a few minor corrections and one request.

In Fig 2A the legend indicates three indentations, but only two are visible. Is the third one obscured or does the third color only apply to panels b and c?

Indeed the placement of the legend was confusing, since in Fig 2A only two indentations are shown. We moved the legend to Fig 2C, and we rewrote the figure caption to make clear that for figure 2A only two indentations are shown. We also slightly adapted the caption for panels b) and c).

I have no idea what is meant by “exponent a of the FDC 1.5,” line 128.

We agree that this sentence was confusing, and we removed it.

Line 238 should refer to Fig 5, not Fig 4, and in line 239, it should refer to Fig. 6.

We thank the reviewer for catching these errors, and we corrected them.

In supplemental Fig. 6A, does ANOVA support the conclusion that all of the patients were significantly different from the healthy donors?

In this work we have compared the donor group against the patient group, which revealed a statistically significant difference between the two groups using a t-test ($p \approx 0.02$). However, we realized that we did not explicitly write in the text that we used a t-test. We thank the reviewer for pointing out this omission. As ANOVA is typically used for more than two populations and t-test in case one only has two populations, we used the latter. Because the reviewer explicitly mentions ANOVA, we also performed – as a check - an ANOVA confirming that the difference between the groups is significant ($p \approx 0.02$). However, we have not compared - and we also do not conclude - that each individual patient has a significantly lower bending modulus than the healthy donors. This is unlikely to be the case. To clarify that our conclusions of a significant difference between the groups is supported by a t-test we now include the following in the figure caption in Fig. 5: “A two-sample t-test revealed that the difference between the donor and patient groups is statistically significant ($p \approx 0.02$).”

Line 267: Supplemental Figure 6

Line 409. There is no Figure 5E.

We thank the reviewer for pointing out these two mistakes; we corrected both.

Given the importance of the assumption that the pressure due to adhesion must be large compared with the pressure increases for small indentations it would be useful to report distribution of the pressures determined from the tethering forces shown in Supplemental figure 5. An example calculation is given in the Supplementary Discussion, but it is not obvious that all of the pressures measured from tethering forces are that high. A simple histogram of the values included as supplemental material would suffice.

We thank the reviewer for this suggestion that helps clarify the point that the initial pressure is large compared to the pressure increase during indentation. We added a new supplemental figure panel to figure S6, in which we present the pressure estimates for the 3 donor and 3 patient samples in box plots. The data indicates that the initial pressure indeed is high for the vast majority of vesicles. We refer to this new figure panel in both the main text and in our supplemental discussion with the example calculation.